# Healing Abutment Distortion in Implant Prostheses: An In Vitro Study

**DOI:** 10.3390/jfb13030085

**Published:** 2022-06-21

**Authors:** Chandrashekhar Pandey, Bishwa Prakash Bhattarai, Apiwat Riddhabhaya, Natthamet Wongsirichat, Dinesh Rokaya

**Affiliations:** 1Department of Clinical Dentistry, International College of Dentistry, Walailak University, Bangkok 10400, Thailand; chandrashekharpandey677@gmail.com (C.P.); bishwa052@gmail.com (B.P.B.); wongsirichatn@gmail.com (N.W.); 2Department of Oral Biosciences, International College of Dentistry, Walailak University, Bangkok 10400, Thailand; atom2497@gmail.com

**Keywords:** dental implants, implant abutment, healing abutment, screw thread, screw head, wear, distortion, scanning electron microscope

## Abstract

Little has been studied regarding the repeated usage of healing abutments and their effects on the distortion of dental implant-healing abutment hex surfaces. Additionally, implant manufacturers do not provide specific guidelines on how many times a healing abutment can be used before discarding. Therefore, we evaluated the effect of repeated screwing-unscrewing of implant-healing abutment on the healing abutment hex surface and screw head. A total of 12 Biomate implants with 4 mm diameter and 13 mm length were inserted into a synthetic bone block. The standard healing abutments of 3 mm diameter and 4 mm length were screwed onto each implant using a torque ratchet at a final torque of 30 Ncm. Immediately, the abutments were unscrewed at 30 Ncm. Then, screwing-unscrewing was repeated for 4, 8, 16, 24, 32, 40, 80, 160, 320, and 400 times and the healing abutments were scanned under the scanning electron microscope for any distortion. Distortion was graded as 0, 1, 2, and 3. Data were analyzed using SPSS 24.0. Descriptive statistics were calculated. One-way ANOVA with post hoc using Tukey’s HSD test was performed to analyze the difference in distortion at different screwing-unscrewing times. A significant level was selected at *p*-value = 0.05. It was found that distortion healing abutments screws were seen after 32 times screwing and unscrewing. There was a significant difference in the distortion (*p*-value < 0.05) after 24 times of repeated usage of healing abutment and at 160, 320, and 400 times. No surface distortions were observed at the healing abutment screw head at 4, 8, 16, 24, 32, 40, 80, 160, 320, and 400 cycles of screwing-unscrewing. It can be concluded that repeated screwing and unscrewing of the implant-healing abutments causes damage to the healing abutment hex surface. The distortion of healing abutments screws was seen after 32 times screwing and unscrewing. No surface distortions were observed on the healing abutment screw head until 400 times of screwing and unscrewing. Hence, the clinician should be cautious while using the healing abutments repeatedly.

## 1. Introduction

Dental implants with digital technologies are widely used for the prosthetic rehabilitation of a missing tooth [1]. A typical dental implant consists of a body, which gets integrated with the native bone, and an abutment, which connects to the implant body and holds the tooth superstructure. Healing abutments and cover screws are interim devices used for special purposes [2].

Healing abutments are temporary components placed on the implant body during the healing period and aid in providing healthy peri-implant mucosa to support the future tooth prosthesis. Implant components are usually manufactured from titanium or its alloys, although newer materials are being tried [3]. The mechanized material is suggested to be a principal factor in increasing the execution of the healing abutment [4,5].

The screw design of the healing abutment and its physical properties vary greatly between manufacturers. The differences in screw design and physical property among healing abutments of different manufacturers may be attributed to the external diameter, thread depth, precision of taper, components, and poor tooling of the healing abutment. These may cause significant differences in complications especially screw loosening [6,7].

Poor connection between healing abutment screw and implant internal hex connection may cause implant-related complications, such as implant-crown dislodgement. Although the interface between the healing abutment screw and implant internal hex connection is important, no studies have been carried out on the biomechanical integrity of the healing abutment screw and implant internal hex connection. Hence, we evaluated the distortion of hex threads of a used healing abutment microscopically [8].

Repeated use of healing abutment may lead to plastic deformation due to repetitive preloads during tightening and retightening. This plastic deformation might cause screw loosening and accidental removal of the abutment in the patient’s oral cavity or might cause micro gaps between the implant-abutment interface [9]. The peri-implant dimensions, as evident from histologic studies, are significantly influenced by the micro gaps at the implant-abutment junction and its location in relation to the alveolar crest, which ultimately affects the peri-implant bone loss [10]. As reported by literature, the mechanism behind this was due to mechanical and biological problems. Biological factors include the formation of bacterial reservoirs near the implant-abutment interface. The mechanical problem of the micro gap results in the micro-movements and possible loosening or fracture of screw-retained abutments [11,12].

Decontamination of implant parts, especially used healing abutments, is important and is achievable through procedures such as autoclave, NOC, air polishing, electrolysis, etc. [13,14,15]. An effective method for decontamination of used healing abutment can be performed by electrolysis by placing the contaminated healing abutments on a carbon cathode and applying the electric current of 1 A at a constant 10 V in 7.5% sodium bicarbonate [14]. This method can be useful in the clinical management of peri-implant infections.

The reusability of healing abutments has often been considered debatable owing to the lack of universal consensus or guidelines. In addition, the implant manufacturers have not provided any recommendations for the repeated usage of their healing abutments, although there is an effect of screwing-unscrewing of healing abutment on the healing abutment hex surface and screw head [16,17]. Hence, the present study was conducted to assess the effect of screwing-unscrewing of implant-healing abutment on the implant hex surface and screw head. The null hypothesis proposes that there is no effect of screwing-unscrewing of healing abutment on the implant hex surface and screw head.

## 2. Materials and Methods

### 2.1. Materials

This study used dental implants and healing abutments, bone block, dental implant torque ratchet (Biomate Medical Devices Technology Co., Ltd., Kaohsiung, Taiwan), and SEM (ProSciTech Pty Ltd., Townsville, Australia). The characteristics of the dental implant parts are summarized in Table 1. The mechanical properties of the implant, healing abutment, and the screwdriver are shown in Table 2.

### 2.2. Implant Insertion

In this study, 12 Biomate implants were used. The sample size calculation was conducted following previous studies in which the sample size ranged from 10 to 15 [21,22]. Implants of size 4 mm in diameter and 14 mm in length were inserted into the synthetic bone block. Three bone blocks were used in our study. Four implants were placed in each bone block at a distance of 5 mm. All the implants were placed equicrestally as per the standard procedure [23]. Copious irrigation was maintained throughout the procedure. Physiodispenser was set at a torque of 40 Newtons and speed of 800 rpm and used to place all the implants as recommendations from previous studies [24,25].

### 2.3. Placement of Healing Abutment

The standard healing abutment (HA) of 3 mm diameter and 4 mm length was used in our study. The 12 healing abutments were divided into two groups; the first group (HA 1 to HA 6) was screwed-unscrewed 4, 16, 32, 80, and 320 times (Figure 1). The second group (HA7 to HA12) was screwed-unscrewed for 8, 24, 40, 160, and 400 times. A torque ratchet at a final torque of 30 Ncm was used for all samples. Then, the abutments were unscrewed, and the procedure was repeated.

One-sided grooves were created on the healing abutment by an ai-rotor handpiece using straight diamond bur for placement, orientation, and identification during the microscopic procedure. Each healing abutment was sent for the SEM evaluation after the aforementioned times of screwing and unscrewing. Two parts of the healing abutments were evaluated under the SEM; first, the connecting screw head of the healing abutment, and second, the screw thread of the healing abutments. The external surface of the healing abutment screw consisting of thread for connection to the corresponding internal hex of the dental implant was hypothesized to deform after repeated loading and unloading. The hollow portion over the superior surface of the screw wherein the screwdriver gets embedded to tighten or loosen the screw from the implant (Figure 2).

### 2.4. Scanning Electron Microscope (SEM)

All the 12 healing abutments were examined using the SEM after repeated screwing and unscrewing. The healing abutments were again scanned after 4, 8, 16, 24, 32, 40, 80, 160, 320, and 400 insertion and removal cycles. Abutment screws were positioned on separate SEM pin-type mounts using conductive adhesive carbon tabs.

The healing abutments were gold-sputtered prior to SEM procedures to make the samples more electroconductive (Figure 3). The hex driver-healing abutment hex interface and screw thread of each healing abutment were analyzed under the SEM, and images were obtained at lower magnifications such that the focus area of each abutment could be visualized to obtain an accurate measurement of the surface distortion [25,26].

### 2.5. Distortion Measurements

The surface distortion of the implant-healing abutment screw (hex thread) was determined under the SEM by using the following criteria using an ordinal scale (Figure 4) [27,28].

Grade 0: Thread surface intact with no visible surface irregularity.

Grade 1: Mild surface irregularities with pitting or slight cracking on the thread surface.

Grade 2: Moderate surface irregularities with deformed thread crests with pitting and slight cracking on the thread surface.

Grade 3: Severe surface irregularities with advanced deformation of thread crests.

The implant-healing abutment screw head distortion was determined under SEM were determined by using the following criteria using an ordinal scale (Figure 5) [27,28].

Grade 0: Screw head intact with no visible surface irregularity.

Grade 1: Mild surface irregularities with pitting or slight cracking on the screw head.

Grade 2: Moderate surface irregularities with deformed screw head with pitting and slight cracking.

Grade 3: Severe surface irregularities with advanced deformation of the screw head.

### 2.6. Statistics Analysis

Data were analyzed using SPSS 24.0 (SPSS, Chicago, IL, USA). Descriptive statistics were calculated. One-way ANOVA with post hoc using Tukey’s HSD test was performed to analyze the difference in distortion at different screwing-unscrewing times. Significance level was selected at *p*-value = 0.05. Only one researcher performed all the research procedures. Then, in the end, all data were verified by random measurements, and data were found valid.

## 3. Result

### 3.1. Healing Abutment Screw Thread

No surface distortions occurred at the healing abutment screw threads up to 16 cycles of screwing-unscrewing (Figure 6). However, after 24, 32, and 40 repeated cycles, mild surface irregularities with pitting or slight cracking on the thread surface of the healing abutment.

The descriptive statistics of surface distortions of the healing abutments after the specific cycles of screwing-unscrewing are shown in Table 3. It was found that distortion healing abutments screws were seen after 32 screwing-unscrewing times.

Multiple comparisons (Table 4 and Table 5) show that there was a significant difference in the distortion (*p*-value < 0.05) after 24 times of repeated usage of the healing abutment.

### 3.2. Healing Abutment Screw Head

No surface distortions were observed at the healing abutment screw head at 4, 8, 16, 24, 32, 40, 80, 160, 320, and 400 cycles of screwing-unscrewing (Figure 7).

## 4. Discussion

We aimed to examine if repetitive screwing and unscrewing could affect the surface characteristics of the healing abutment. This is the first study to investigate the same to the best of our knowledge. The null hypothesis is rejected, and there are some surface changes following the screwing-unscrewing of healing abutment on the implant hex surface and screw head. Various methods to measure the surface wear are the 2D surface profilometry [29,30], SEM [31,32], 3D images superimposition using computer software [33,34], wear volume calculation using 3D images from focus variation microscopy [30] and wear particle characterization [35]. Stress distribution analysis can be performed from the finite element (FE) study [36]. However, in our study, 3D image superimposition was not a suitable method as the 3D scan images using Trios 3 (3Shape Dental Systems, Copenhagen, Denmark) were not of adequate quality due to the small size of the healing abutment hex, and the 3D scan cannot capture the hex threads. This is similar to the case of the light profilometer. Due to the hex thread shape, contact profilometers were also not suitable. Finally, wear volume and wear particle characterization were also not suitable due to the implant-healing abutment connection. Hence, we used SEM to analyze the wear of the healing abutment hex.

From this study, the microscopic pictographs confirmed this at higher resolution taken after recurring cycles, in which we observed thread and hex driver insertion surface distortion of healing abutments. We also found that the abutment removal torque gradually lowed than the insertion torque with increased repetitive screw unscrew cycles. These results could be explained based on the behavior of prosthesis abutment and hence would be discussed accordingly henceforth.

The friction helps in the retention of the restoration, contributes the joint stability, and resists screw loosening. Increasing screw insertion cycles decreases the friction between the screw and internal threads of the implant. We observed that after 24, 32, and 40 times repeated cycling of loosening and tightening of healing abutment causes mild surface irregularities with pitting or slight cracking on the thread surface of healing abutment occurring in all the tested healing abutments. This has been confirmed by SEM analysis that increasing screw closure cycles caused changes in the surface of the screw. The clinical implications of the screwing and unscrewing are shown in Table 6. It is not recommended to use the same healing abutment after 8–10 patients.

Our findings were in concordance with the results of Delben et al. [16], Carsdosa et al. [17], and Arshad et al. [23], who observed screw loosening and reduced torque after cyclic mechanical retightening. This could be due to the mechanical galling of metal surfaces due to their repetitive adherence to each other.

Another factor responsible could be the brittleness of titanium screws due to the body-centric cubic crystal structure making them more vulnerable to distortion [17]. Our findings further confirm those of Weiss et al. [25], Ricciardi Coppedê et al. [26], and Kim et al. [27], who also demonstrated that frequent mechanical retightening causes retention loss of stability torque.

Assunção et al. [37] studied the maintenance of tightening torque in various retention screw types of implant crowns and found that all screws exhibited a reduction in the detorque. The Ti screw showed the highest torque, whereas the gold-coated screw and the Ti screw with TiAlN coating showed the lowest torque. Similarly, Pardal-Peláez and Montero [38] studied the preload loss of the abutment screws in internal and external connections of dental implants, and they mentioned that internal connection with morse taper resisted cyclic loading of screw loosening.

The coefficient of friction among two metals is maintained by their intrinsic metallurgic properties and the mechanized process, which determines their geometric design and surface texture, which might be the reason previous studies observed differences in pre-facture torque value of abutments from different brands and within the same brand, and different shipments as well [24,25].

In a clinical setting, surface distortion could lead to the formation of micro gaps, causing microleakage of bacterial substrates or accidental slippage of the abutment in the oral cavity, which might lead to its aspiration [39,40].

The mechanical properties of the implant and the screwdriver (Table 2) can affect the distortion of the healing abutment due to the friction while screwing and unscrewing the healing abutment. In our study, the implants and healing abutments have similar mechanical properties as they were made from similar materials (titanium grade 4), whereas the mechanical properties of the screwdriver were lower mechanical as they were made from stainless steel. Hence, we assumed that the implant and screwdriver have no or very less effect on the distortion of the healing abutment hex and screw head.

This study can be extended to study the wear by advanced imaging methods in the future. In addition, the type of stress distribution can be studied using the FE study. The in vitro study design of the present study is one of its potential limitations as it could not simulate the exact biologic surroundings of the oral cavity. Thus, the findings should be validated in a clinical condition. The decontamination (autoclaving) of the healing abutments is performed in between repeated use [13], but we did not do the decontamination as this study is an in vitro study. Moreover, it has been shown that in SEM analysis, there were no detectable differences in the surface of Ti healing abutments the following decontamination using autoclaving, chemicals, or air polishing. In addition, only a single implant brand has been tested in our study, which could not justify the behavior of other implant systems. Finally, in this study, multiple screw insertion cycles and constant friction were not assumed. We could not perform this study in more samples due to the limitations of the research budget and time. In further studies, proper design, the passive fit of the prosthesis, and the correct patient selection are vital, controlling and treating the possible para functions and occlusal overloads.

## 5. Conclusions

Within the limitation of this study, it can be concluded that repeated screwing and unscrewing of the implant-healing abutments causes damage to the healing abutment hex surface. The distortion of healing abutments screws was seen after 32 times screwing and unscrewing. No surface distortions were observed on the healing abutment screw head until 400 times of screwing and unscrewing. Hence, the clinician should be cautious while using the healing abutments repeatedly.

## Figures and Tables

**Figure 1 jfb-13-00085-f001:**
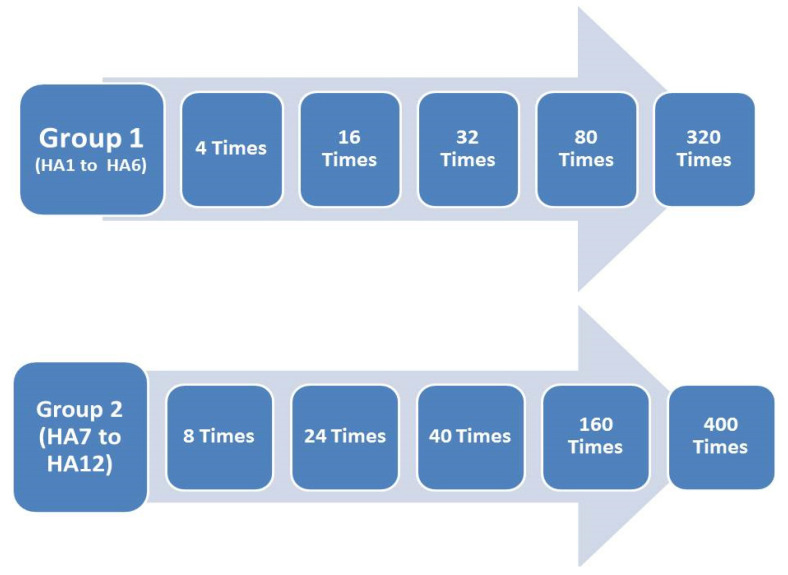
Study groups of screwing-unscrewing used in this study.

**Figure 2 jfb-13-00085-f002:**
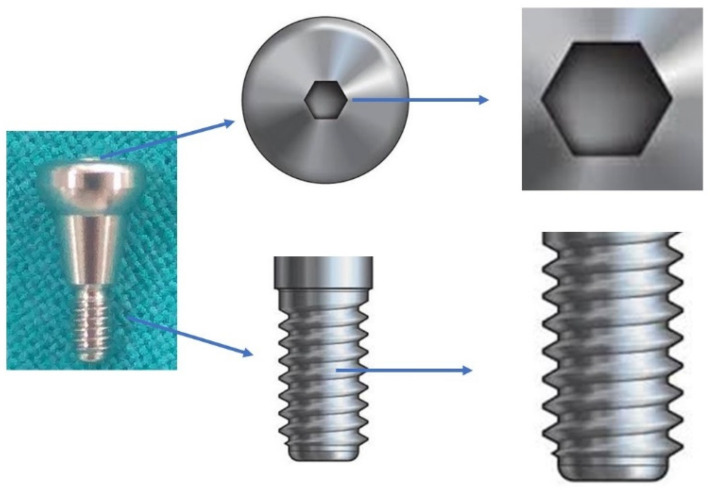
Healing abutment hex surface and screw head.

**Figure 3 jfb-13-00085-f003:**
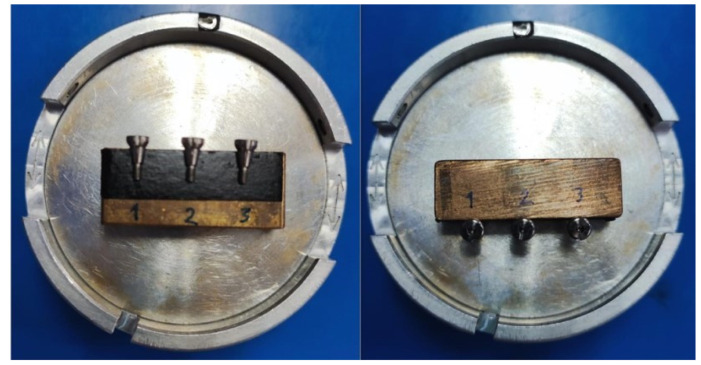
Sample preparation for the scanning electron microscope (SEM).

**Figure 4 jfb-13-00085-f004:**
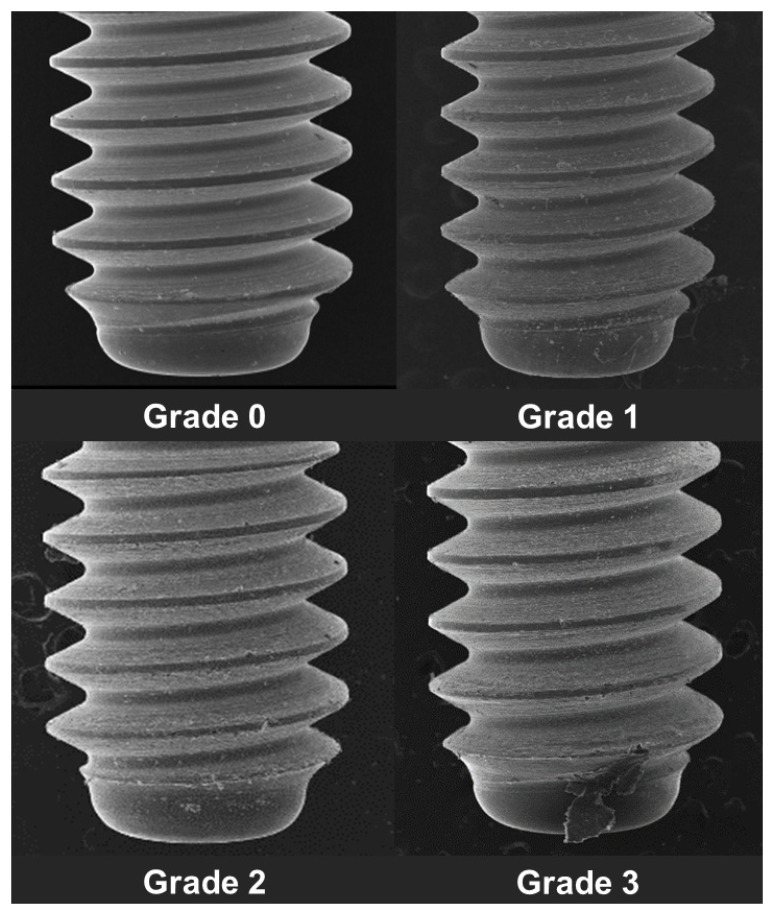
Grading of implant-healing abutment hex surface distortion.

**Figure 5 jfb-13-00085-f005:**
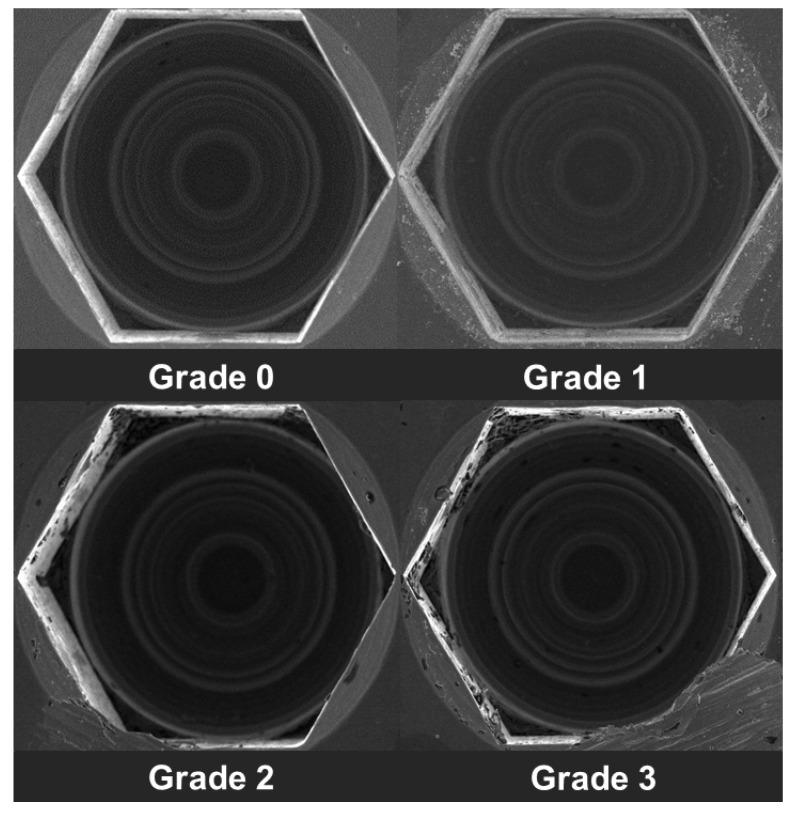
Grading of implant-healing abutment screw head distortion.

**Figure 6 jfb-13-00085-f006:**
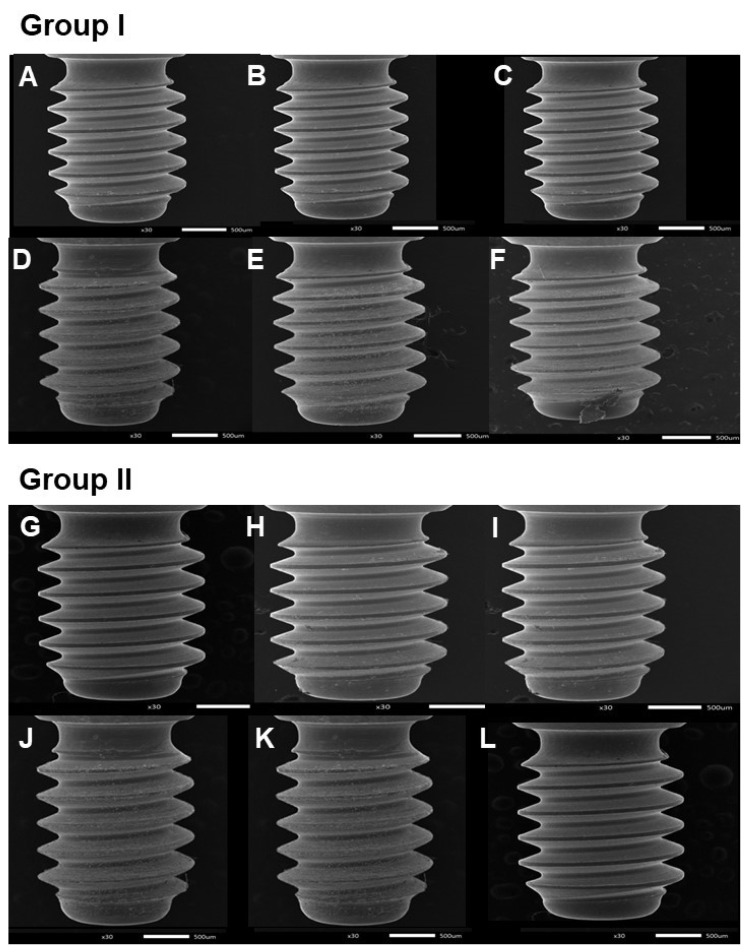
Scanning electron microscope (SEM) images of healing abutment screw. Group I; (**A**) = before screwing-unscrewing, (**B**) = after 4 times of screwing-unscrewing, (**C**) = after 16 times of screwing-unscrewing, (**D**) = after 32 times of screwing-unscrewing, (**E**) = after 80 times of screwing-unscrewing, (**F**) = after 320 times of screwing-unscrewing. Group II; (**G**) = before screwing-unscrewing, (**H**) = after 8 times of screwing-unscrewing, (**I**) = after 24 times of screwing-unscrewing, (**J**) = after 40 times of screwing-unscrewing, (**K**) = after 160 times of screwing-unscrewing, and (**L**) = after 400 times of screwing-unscrewing.

**Figure 7 jfb-13-00085-f007:**
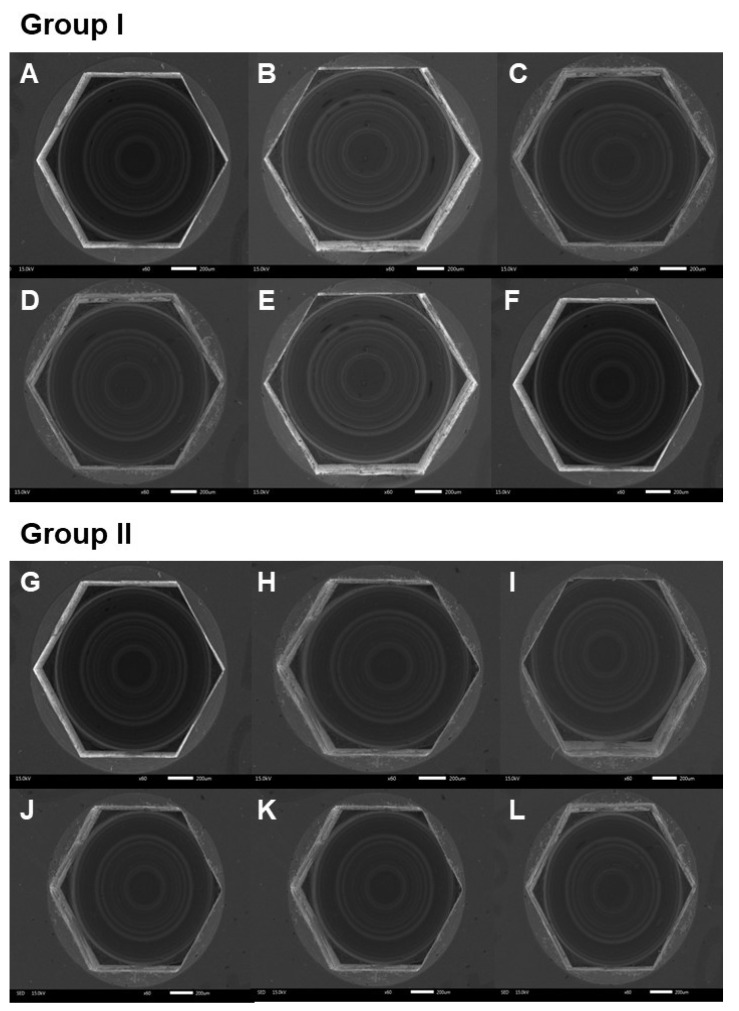
Scanning electron microscope (SEM) images of healing abutment screw head. Group I; (**A**) = before screwing-unscrewing, (**B**) = after 4 times of screwing-unscrewing, (**C**) = after 16 times of screwing-unscrewing, (**D**) = after 32 times of screwing-unscrewing, (**E**) = after 80 times of screwing-unscrewing, (**F**) = after 320 times of screwing-unscrewing. Group II; (**G**) = before screwing-unscrewing, (**H**) = after 8 times of screwing-unscrewing, (**I**) = after 24 times of screwing-unscrewing, (**J**) = after 40 times of screwing-unscrewing, (**K**) = after 160 times of screwing-unscrewing, and (**L**) = after 400 times of screwing-unscrewing.

**Table 1 jfb-13-00085-t001:** Characteristics of the dental implant used in this study.

Diameter	4 mm
Length	14 mm
Type	Bone type implant system
Surface treatment	Laser-etched
Titanium type	Grade IV
Hex type	Internal hex with morse taper connection

**Table 2 jfb-13-00085-t002:** The mechanical properties of the dental implant, healing abutment, and screwdriver [18,19,20].

Properties	Implant	Healing Abutment	Screw Driver Ratchet
Materials	Titanium Grade 4	Titanium Grade 4	Stainless steel
Tensile strength (MPa)	550	550	520–670
Yield strength (0.2% offset; MPa)	485	485	190
Elongation (%)	15	15	45 Min %
Poisson’s ratio	0.37	0.37	0.265
Density	4.51 g/cm^3^	4.51 g/cm^3^	8.00 g/cm^3^

**Table 3 jfb-13-00085-t003:** Descriptive statistics of the mean scores of surface distortions of the healing abutments screw threads after the specific cycles of screwing-unscrewing.

Groups	Mean Score	Standard Deviation	Standard Error
Before	0.00	0.000	0.000
4	0.00	0.000	0.000
8	0.00	0.000	0.000
16	0.00	0.000	0.000
24	1.00	0.000	0.000
32	2.00	0.000	0.000
40	2.00	0.000	0.000
80	2.00	0.000	0.000
160	2.17	0.408	0.167
320	2.17	0.408	0.167
400	2.33	0.516	0.211

**Table 4 jfb-13-00085-t004:** Multiple comparisons of the surface distortions of the healing abutments screw threads after screwing-unscrewing in Group I (4, 16, 32, 80, 320).

(I) Groups	(J) Groups	Mean Difference (I−J)	Std. Error	*p*-Value
Before	4	0.000	0.135	1.000
	16	0.000	0.135	1.000
	32	−2.000 *	0.135	<0.0001 *
	80	−2.000 *	0.135	<0.0001 *
	320	−2.167 *	0.135	<0.0001 *
4	16	0.000	0.135	1.000
	32	−2.000 *	0.135	<0.0001 *
	80	−2.000 *	0.135	<0.0001 *
	320	−2.167 *	0.135	<0.0001 *
16	32	−2.000 *	0.135	<0.0001 *
	80	−2.000 *	0.135	<0.0001 *
	320	−2.167 *	0.135	<0.0001 *
32	80	0.000	0.135	1.000
	320	−0.167	0.135	0.975
80	320	−0.167	0.135	0.975

* Significant difference at *p*-value < 0.05. Post hoc using Tukey’s HSD test.

**Table 5 jfb-13-00085-t005:** Multiple comparisons of the surface distortions of the healing abutments screw threads after screwing-unscrewing in Group II (8, 24, 40, 160, 400).

(I) Groups	(J) Groups	Mean Difference (I−J)	Standard Error	*p*-Value
Before	8	0.000	0.135	1.000
	24	−1.000 *	0.135	<0.0001 *
	40	−2.000 *	0.135	<0.0001 *
	160	−2.167 *	0.135	<0.0001 *
	400	−2.333 *	0.135	<0.0001 *
8	40	−2.000 *	0.135	<0.0001 *
	160	−2.167 *	0.135	<0.0001 *
	320	−2.167 *	0.135	<0.0001 *
	400	−2.333 *	0.135	<0.0001 *
24	40	−1.000 *	0.135	<0.0001 *
	80	−1.000 *	0.135	<0.0001 *
	160	−1.167 *	0.135	<0.0001 *
	400	−1.333 *	0.135	<0.0001 *
40	160	−0.167	0.135	0.975
	400	−0.333	0.135	0.343
160	400	−0.167	0.135	0.975

* Significant difference at *p*-value < 0.05. Post hoc using Tukey’s HSD test.

**Table 6 jfb-13-00085-t006:** The clinical implications of the times of screwing and unscrewing in patients.

Times of Screwing and Unscrewing the Healing Abutment	Number of Patients
4	1 patient
8	2 patients
16	4 patients
24	6 patients
32	8 patients
40	10 patients
80	20 patients
160	40 patients
320	80 patients
400	100 patients

## Data Availability

Not applicable.

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
