# Peer review of "Healing Abutment Distortion in Implant Prostheses: An In Vitro Study"

_jfb, 2022, doi:10.3390/jfb13030085_

Round 1

Reviewer 1 Report

The manuscript describes the in-vitro Healing Abutment Distortion of Implant Prostheses. The topic is interesting and has important clinical implications. However, I would like to suggest to improve introduction and references on the topic. 

Author Response

Response to Reviewer 1 Comments

The manuscript describes the in-vitro Healing Abutment Distortion of Implant Prostheses. The topic is interesting and has important clinical implications. However, I would like to suggest to improve introduction and references on the topic.

Thank you for your positive comments. All your comments have been addressed. Corrections in the Manuscript are highlighted in pink color.

Response: Introduction and the References have been improved.

Reviewer 2 Report

Reviewer comments jfb-1703536

General:

This is an interesting study investigating a relevant area in implant Prosthodontics. The study has been conducted well, however a major shortcoming with the study is the ‘design’, in which no provision was in place to replicate the decontamination (autoclaving) of the healing abutments in between repeated use (Please refer related article- Chew et al, Reusing Titanium Healing Abutments: Comparison of Two Decontamination Methods Int J Prosthodont 2018;31:613–618. doi: 10.11607/ijp.5881. There are others in the literature too discussing this aspect). So, consequently, the clinical significance of the study becomes somewhat questionable. The study at best shows some surface topographical aspects of the healing abutments after repeated use (with SEM analysis) without scope for extrapolation to clinical situations.  

Abstract:

Introduction/ Background:

- Page 2, lines 54-55: ‘Hence, we evaluated the distortion of hex threads of a used healing abutment microscopically’.

This is valid only if decontamination cycles are incorporated after each use as they might significantly influence the results, including the screwhead contour affecting the engagement of the screwdriver and effective tightening and loosening.

- The introduction needs to deal with the disinfection of healing abutments (is currently missing) and present the earlier results on the surface changes of leaking abutment after the procedures. Also, provide reasons why this was not done in the current study and how it would potentially affect the results.

Materials & Methods:

- Page 3, lines 92-93: Why were the healing abutments not just hand tightened but at 30Ncm. Is this clinically relevant. Please elaborate in the discussion section?

- Intra-operator or inter-operator variability needs to be mentioned aothougb there is a standard torque used to tighten the abutments.

Discussion:

- These results are specific to a particular system and this needs to be emphasized.

Conclusions:

The conclusions need to be amended keeping in mind the limitation of the study. 

Author Response

Response to Reviewer 2 Comments

Thank you for your positive comments. All your comments have been addressed. Corrections in the Manuscript are highlighted in Yellow color.

This is an interesting study investigating a relevant area in implant Prosthodontics. The study has been conducted well, however a major shortcoming with the study is the ‘design’, in which no provision was in place to replicate the decontamination (autoclaving) of the healing abutments in between repeated use (Please refer related article- Chew et al, Reusing Titanium Healing Abutments: Comparison of Two Decontamination Methods Int J Prosthodont 2018;31:613–618. doi: 10.11607/ijp.5881. There are others in the literature too discussing this aspect). So, consequently, the clinical significance of the study becomes somewhat questionable. The study at best shows some surface topographical aspects of the healing abutments after repeated use (with SEM analysis) without scope for extrapolation to clinical situations.

Response: In this study, we evaluated the effect of repeated screwing-unscrewing of healing abutment on the implant hex surface. We didn’t do the decontamination (autoclaving) of the healing abutments in between repeated use as the study was done on the study model. We have added this at the end of the Discussion with the above reference.

In addition, this can be done clinically in more subjects. Added in the Discussion.

Abstract. Introduction/ Background:

- Page 2, lines 54-55: ‘Hence, we evaluated the distortion of hex threads of a used healing abutment microscopically’. This is valid only if decontamination cycles are incorporated after each use as they might significantly influence the results, including the screwhead contour affecting the engagement of the screwdriver and effective tightening and loosening.

- The introduction needs to deal with the disinfection of healing abutments (is currently missing) and present the earlier results on the surface changes of leaking abutment after the procedures. Also, provide reasons why this was not done in the current study and how it would potentially affect the results.

Response: In this study, we evaluated the effect of repeated screwing-unscrewing of healing abutment on the implant hex surface. We didn’t do the decontamination (autoclaving) of the healing abutments in between repeated use as the study was done on study model. We have added this at the end of the Discussion.

The disinfection (autoclaving) of healing abutments is added in the Introduction.

Materials & Methods:

- Page 3, lines 92-93: Why were the healing abutments not just hand tightened but at 30 Ncm. Is this clinically relevant. Please elaborate in the discussion section?

- Intra-operator or inter-operator variability needs to be mentioned although there is a standard torque used to tighten the abutments.

Response: Line 100-101. A torque ratchet at a final torque of 30 Ncm was used for all samples.

Only one researcher performed all the research procedure. And the at the end all data were verified by random measurements and data were found valid. This is added on the Method.

Discussion:

- These results are specific to a particular system and this needs to be emphasized.

Response: In this research, we used only one system. This limitation is added in the Discussion.

Conclusions:

The conclusions need to be amended keeping in mind the limitation of the study. 

Response: Conclusion is edited and improved.

Reviewer 3 Report

Dear Editor,

Regarding the submitted manuscript “   Healing Abutment Distortion of Implant Prostheses: An in-vitro Study” the presented study is intended to be an in vitro study to assess the healing abutment distortion.

Overall appreciation

I think the authors should address some corrections/clarifications and resubmit:

1-Hypothesis – The authors need to state the null hypothesis

2-Was the sample size adequate? – I cannot find any sample size calculation or a power analysis.  To address the sample size the authors, need to state what would be the mean differences to address with the proposed power.

5-In the material and methods section a professional statistical analysis is advised since I believe that dichotomic analysis could be performed. Additionally, I believe that the sample size is rather small.

6-Discussion:

  1. The authors should start by reporting if the proposed hypothesis was rejected or accepted.
  2. The external validity should be better clarified since being an in vitro study, extrapolation should be performed with caution since biological behavior is different.

Based on the manuscript analysis I believe that the manuscript should be considered for publication after reformulation.

Author Response

Response to Reviewer 3 Comments

Regarding the submitted manuscript “Healing Abutment Distortion of Implant Prostheses: An in-vitro Study” the presented study is intended to be an in vitro study to assess the healing abutment distortion. Overall appreciation. I think the authors should address some corrections/clarifications and resubmit:

Thank you for your positive comments. All your comments have been addressed. Corrections in the Manuscript are highlighted in green color.

1-Hypothesis – The authors need to state the null hypothesis

Response: Hypothesis is added in the Introduction and explained in the Discussion.

2-Was the sample size adequate? – I cannot find any sample size calculation or a power analysis.  To address the sample size the authors, need to state what would be the mean differences to address with the proposed power.

Response: Regarding the sample size selected, we have added in the Method.

In this study, we used 12 dental implants. And the screwing and unscrewing are done in till 600 times. For such in-vitro studies in dental implants, it is expensive to purchase dental implants, so such studies are generally done in around 10-15 studies as follows.

https://www.hindawi.com/journals/ijbm/2012/181024/

https://www.hindawi.com/journals/bmri/2021/3582342/

The limitation of the sample size is added in the limitation also.

5-In the material and methods section a professional statistical analysis is advised since I believe that dichotomic analysis could be performed. Additionally, I believe that the sample size is rather small.

Response: Statistical analysis is done by consultation with a statistician and One-way ANOVA is used as the comparison is done in 6 groups. We used references from other studies. https://www.hindawi.com/journals/ijbm/2012/181024/

https://www.hindawi.com/journals/bmri/2021/3582342/

We couldn’t perform this study in more samples due to limitations of the research budget and time. We have mentioned this in the limitation of this study.

6-Discussion:

  1. The authors should start by reporting if the proposed hypothesis was rejected or accepted.
  2. The external validity should be better clarified since being an in vitro study, extrapolation should be performed with caution since biological behavior is different.

Based on the manuscript analysis I believe that the manuscript should be considered for publication after reformulation.

Response: The hypothesis is explained in the Discussion. The external validity of this invitro study is clarified. Line 200-205 and 241-242.

Round 2

Reviewer 3 Report

From the comments there are two major points that need adressing:

1- Sample size calculation and power analysis - sample size calculation is based in effect level and power analysis accordding to what the authors need to compare. You cannot use other studies with different variables to adress the sample size. Sample size calculation is a mathematical issue as should be adressed as such.

2- Statistical support - I believe that professional statistical support is needed since sample size, power analysis, effect size or a normality check is something every statistician knows how to adress.

Due to the small sample size it is this reviewer opinion that the points mentioned need to be correctly adressed before considering for  publication.

Author Response

Response to Reviewer 2 Comments

From the comments there are two major points that need adressing:

Thank you for your positive comments. All your comments have been addressed. Corrections in the Manuscript are highlighted in Yellow color.

1- Sample size calculation and power analysis - sample size calculation is based in effect level and power analysis accordding to what the authors need to compare. You cannot use other studies with different variables to adress the sample size. Sample size calculation is a mathematical issue as should be adressed as such.

2- Statistical support - I believe that professional statistical support is needed since sample size, power analysis, effect size or a normality check is something every statistician knows how to adress.

Due to the small sample size it is this reviewer opinion that the points mentioned need to be correctly adressed before considering for publication.

Response: In this study, we did not calculate the sample size. We used 12 dental implants. And the screwing and unscrewing are done in till 600 times. We had to limit the size to 12 dental implants due to the costs of the dental implants. Statistical support was obtained. The limitation of the sample size is added to the limitation also.